# The Key to Solving Plastic Packaging Wastes: Design for Recycling and Recycling Technology

**DOI:** 10.3390/polym15061485

**Published:** 2023-03-16

**Authors:** Qian Ding, Heping Zhu

**Affiliations:** China Packaging Design and Innovation Center, School of Packaging Design and Art, Hunan University of Technology, Zhuzhou 412007, China

**Keywords:** plastic packaging, pollution treatment, design for recycling, recycling technology, circular economy

## Abstract

Confronted with serious environmental problems caused by the growing mountains of plastic packaging waste, the prevention and control of plastic waste has become a major concern for most countries. In addition to the recycling of plastic wastes, design for recycling can effectively prevent plastic packaging from turning into solid waste at the source. The reasons are that the design for recycling can extend the life cycle of plastic packaging and increase the recycling values of plastic waste; moreover, recycling technologies are helpful for improving the properties of recycled plastics and expanding the application market for recycled materials. This review systematically discussed the present theory, practice, strategies, and methods of design for recycling plastic packaging and extracted valuable advanced design ideas and successful cases. Furthermore, the development status of automatic sorting methods, mechanical recycling of individual and mixed plastic waste, as well as chemical recycling of thermoplastic and thermosetting plastic waste, were comprehensively summarized. The combination of the front-end design for recycling and the back-end recycling technologies can accelerate the transformation of the plastic packaging industry from an unsustainable model to an economic cycle model and then achieve the unity of economic, ecological, and social benefits.

## 1. Introduction

Nowadays, plastics have become one of the most important fundamental materials and are widely applied in agricultural production, medical treatment, electrical and electronics, packaging, transportation, aerospace, and other fields [1,2,3,4,5,6,7]. Note that packaging is the largest application field for plastics, which accounts for about 40% of the global total yield of plastics [8]. Plastics are often served as single-use products due to their good performance, such as their light weight, low cost, easy processing, and excellent mechanical properties [9]. But at the same time, the degradation rates of most plastics in the environment are extremely slow because of their stable chemical structure [10,11]. Therefore, it is conceivable that a great deal of waste plastic is generated every year. A recent report by the Organization for Economic Co-operation and Development (OECD) demonstrated that the production of plastics has increased from 2 million tons in 1950 to 460 million tons in 2019. However, only 8% of plastic waste was recycled, and almost 50% of plastic waste was landfilled [12]. Since the COVID-19 pandemic, people’s demands for anti-epidemic supplies have gradually increased; meanwhile, the amounts of online shopping and delivery have exploded [13,14,15]. As a result, the plastic waste problem is worsening dramatically.

The best management of plastic waste is undoubtedly a major challenge for the world, and controlling plastic waste has become a global consensus [16]. In the last two decades, plastic waste treatment and pollution control have become the theme of World Environment Day several times, while governments have spared no effort to reduce the use of single-use plastic products and improve the recycling of plastic waste [17,18,19]. For instance, China released “Opinions on Further Strengthening the Control of Plastic Pollution” in 2020, which prohibits the production and sale of ultra-thin plastic shopping bags with a thickness of less than 0.025 mm and polypropylene (PE) agricultural mulch film with a thickness of less than 0.01 mm [20]. However, the treatment of plastic waste is not satisfactory due to the high recycling cost, unresolved technical problems, and inappropriate consumption habits [21,22]. Recently, it was proposed that the nature of plastic waste is improper management, which causes the leakage of plastic waste into the environment [23]. Thus, plastics are not pollutants and can be turned into valuable resources through reasonable and effective treatment [24].

Significantly, the resolution titled “End plastic pollution: towards a legally binding instrument” was approved at the fifth session of the United Nations Environment Assembly (UNEA-5.2) in 2022, which is the first legal document for the United Nations to tackle plastic waste [25]. The above resolution clearly proposed the concept of lifecycle management of plastics, including design, production, consumption, and recycling stages [26,27,28,29,30]. Firstly, reuse and recycling of plastic products are regarded as important considerations during the design phase. Secondly, the idea of circular economics should be included in the production process, and a variety of recyclable plastic products should be encouraged to be produced through technical innovation. Thirdly, governments should promulgate relevant laws and regulations to urge consumers to change their consumption habits, thereby reducing the usage of plastic products. Finally, degrading enzymes can be added to accelerate the degradation of plastic waste; furthermore, prefabricated panels, fuel, and recycled plastic products can be prepared from plastic waste.

In a scientific view, the best way to deal with plastic waste is through the recycling of plastic waste, which contributes to realizing the closed-loop circulation of plastics [31], as shown in Figure 1. However, there is a wide variety of plastic products with different properties, resulting in difficult treatment, a high cost, and a low recovery rate [32,33,34]. According to different sources, plastic wastes can be classified into industrial, agricultural, medical, and household plastic wastes [35,36,37]. In general, industrial plastic wastes have a high recovery value due to their clear source and good quality, while the recycling of agricultural and medical plastic wastes is extremely difficult because of their poor qualities, wide dispersion, and direct or indirect infectious, toxic, and other hazards. As to household plastic wastes, especially plastic packaging wastes, their recycling rate has strong growth potential in the coming years, largely due to domestic waste classification [38,39]. Although companies and researchers pay much attention to the back-end recycling of plastic packaging wastes, the front-end design for recycling plastic packaging is ignored.

It is known that design is the beginning of the lifecycle of plastic products; therefore, the service period of plastic products can be lengthened by improving their recoverability [40,41]. In reality, the front-end design for recycling determines 80–90% of the recovery value of plastic packaging; that is, the recovery rate of plastic packaging wastes and quality of recycled products depend not only on the recycling technologies but also on the front-end design for recycling. This review emphasized that plastic waste should be prevented and controlled throughout the life-cycle of plastic products, covering design, production, utilization, sorting, and recycling. Besides, all-round countermeasures for plastic wastes were proposed through comprehensive analysis of the front-end design for recycling and the back-end recycle and reutilization technology of plastic packaging.

## 2. Design for Recycling of Plastic Packaging

### 2.1. Design for Recycling

The concept of design for recycling appeared in the 1990s, and its early applications were mainly in the fields of automobiles, furniture, and electric and electronic products [42,43,44,45]. For example, IBM recycled over 70 million pounds of computer equipment from 1994 to 1997, saving over $50 million through internal product reuse and bringing in another $5 million through the sale of recycled commodities [46]. Up to now, much literature has proposed the principles of recyclability, for example, by reducing the diversity of materials, adopting easy-to-assemble and disassemble structures, and selecting recyclable materials [47,48,49,50,51].

Design for recycling refers to a series of design ideas and methods that fully consider and solve problems related to the recyclability of products, such as recycling possibility, recovery value, recovery methods, and the recovery treatment technique of part materials, and therefore improve the recycling rate of parts and components [52,53,54]. As an important part of green design, planning for recycling can not only reduce plastic emissions but also increase the recycling of plastic waste. The main contents of the design for recycling are as follows [55,56,57,58,59,60].

(1)Recyclable materials and marks

The recycling possibility of discarded products depends on their own material properties as well as property retention; therefore, product designers need to understand the variation of properties of materials [61]. Meanwhile, in view of the fact that not all parts of the product are recyclable, labeling of parts for the type of material and marking of the recyclable components are necessary at the design stage [62].

(2)Recovery techniques and methods

The recycling methods for parts and materials from scrapped products are different. Some of these materials are able to be reused directly, while others require different treatments before being recycled [63]. Thus, product designers need to understand and grasp different methods of recycling. Note that the technique and methods for product recovery are not merely formulated by the design department but developed by different research departments and then shared with the department.

(3)Recycling economy

The recycling economy is the determining factor for material recycling, which requires product designers to master the economy of recycling and understand the real-time conditions of the recycling market. Referring to the current cost budgeting method, a series of evaluation models have been built through collecting and analyzing related data during the design and manufacture processes [64,65].

(4)Structure and design of recyclable parts

The primary condition of recycling is to disassemble parts from products easily, undamaged, and at a reasonable cost, which means that the structure of products has to be designed for convenient disassembly. For example, the parts of products can be designed as an organization of functional units that are easily accessible, easy to assemble, and easy to separate [66].

Although design for recycling has been widely used in the field of product manufacturing, there is little application in packaging, particularly plastic packaging. As early as 1988, Selke et al. [55] discussed the recyclability of different packaging materials, such as metals, glass, paper and paperboard, plastics, and multi-layer composites. They pointed out that plastics are less recyclable than metal and glass because of their decreased chemical and physical properties, as well as inevitable contamination. In addition, there is a large variety of plastics, such as polyethylene (PE), polypropylene (PP), polystyrene (PS), polyvinyl chloride (PVC), polyethylene terephthalate (PET), polylactic acid (PLA), and so on, which results in a complicated and high-cost separation and recovery process. In recent years, with the growing problem of plastic waste, the demands for design for recycling and recycling certification for plastic packaging have grown more than ever before.

### 2.2. Recyclability and Influencing Factors

A clear definition of recyclability is essential to assessing whether the design of plastic packaging is recyclable or not. Thus, a global definition of “recyclability” for plastic packaging and products was put forward in 2018 [67]. That is, plastic products that meet the following four conditions are considered recyclable. First, the product must be made of plastic that is collected for recycling, has a market value, and/or is supported by a legislatively mandated program; secondly, the product must be sorted and aggregated into defined streams for recycling processes; thirdly, the product can be processed and reclaimed or recycled with commercial recycling processes; and lastly, the recycled plastics become raw materials that are used in the production of new products. Although the above definition is generally accepted, the evaluation and testing standards for recyclability are only carried out in a few countries or regions. For instance, the Association of Plastic Recyclers (APR) in America has developed a series of design guides, which are accepted by most US plastic manufacturers and brands [68]. Moreover, the design for recycling needs to be compatible with local collection and recycling systems [69].

The design for recycling plastic packaging is a huge system that includes related standards for individual plastics and different application areas, such as rigid plastic containers and flexible packaging. Here, the “PET Products Recyclability Design Guidance” is taken as an example to illustrate how to solve practical problems of plastic recycling. It is known that PET bottles have one of the highest recovery rates among plastic packaging. As reported, PET bottles account for 62% of produced bottles, and the recycling rate of PET bottles is about 50% worldwide [70]. However, the recycled PET bottles are used to produce low-value products such as strapping, sheeting, and fiber instead of new bottles. To achieve bottle-to-bottle recycling of PET, it requires a full understanding of design elements affecting recyclability, which include base resin, color and dimensions, barrier layer/coating/additives, closures and pumps, labels/inks/adhesive, and attachments [68,71].

(1)Base resin

PET copolymer resins, having a crystalline melting point between 225 and 255 °C, bio-based PET resins, as well as recycled PET (r-PET), have good compatibility with PET and thus are preferred designs. Nevertheless, blends of PET and other plastics need to be tested to evaluate their recyclability, and the resins that are not compatible with PET should be avoided in the beginning.

(2)Barrier layer/coating/additives

In order to meet specific packaging performance requirements, polyvinyl dichloride (PVDC), polyamides (PA), silicon oxide coatings, nucleating agents, optical brighteners, and other additives are added to PET; however, some of these additives have a negative effect on the recycling process of PET. For instance, PVDC has excellent barrier properties and is often used as the barrier layer of bottles [72]. But the melting temperature of PVDC is much lower than that of PET, and then thermal degradation of PVDC occurs inevitably during the melting processing of PET. Therefore, the influence of these non-PET materials on the recycling process and properties of r-PET needs to be considered and tested in the material design stage.

(3)Labels/inks/adhesive

In general, labels and inks are necessary parts of packaging and play a role in disseminating information about products. Label films are usually pasted on the surface of packaging or wrapped around the bottle surface through thermal shrinkage. Although labels and inks represent a small part of packaging, they have a great impact on the recovery process of PET. As the PET bottles are accurately identified by near-infrared (NIR) spectrum sorting machinery even with the label attached, the adhesives should be thoroughly removed from the bottles. According to the RECOUP Recommendations [73], the proposed label size is no more than 40% of the surface area of bottles in order to ensure the accuracy of the automated sorting process of PET bottles. In addition, non-staining label inks are suggested to avoid discoloring the PET flake.

(4)Closures and pumps

The density of PET is 1.38 g/cm^3^, so plastics with densities less than 1 g/cm^3^ are preferred to produce closures and pumps. Due to the density difference, closures and pumps are able to be separated from PET in the flotation process. Although some kinds of plastics with densities higher than 1 g/cm^3^ can be removed through NIR sortation or melt filtration, these materials have a negative effect on the reprocessing of PET and need to be improved. Furthermore, the metal spring, stainless steel beads, or glass beads should be replaced by PP or PE.

(5)Color and dimensions

Unpigmented PET, transparent light blue, and light green PET have a high cost effectiveness and are possible to be processed into value-added products, while opaque colors probably cause contamination in the PET recycling feed stream. Besides, the production of white-colored PET bottles has to be avoided because they are not separated from the resin and then decrease the recycling value of r-PET. The dimensions of plastic items range from 5 cm to 7.5 L, which is preferred because items with too small or too large size are not accepted or distinguished by automatic sorting equipment.

(6)Attachments

Normally, attachments made of PET or easily separated are preferred. Paper attachments are rendered into a pulp during the caustic wash process and are difficult to filter from the liquid. PVC and PLA have similar densities to PET, which is not possible to remove by means of the sink-float separation process. In addition, welded attachments and RFIDs (radio frequency identification devices) are not compatible with PET and may bring about processing problems [74].

### 2.3. Application Cases and Strategies

In recent years, many well-known brands have made public commitments on plastic packaging recycling and taken action to improve the design of plastic packaging [75,76]. For example, Coca-Cola, Pepsi, and Nestle promised to achieve 100% recyclability or reuse of their packaging by 2025. Table 1 summarized some representative design cases of plastic packaging, which provide references for other packaging producers.

It is widely known that the basic functions of packaging are to protect products, make them convenient to use, and promote sales. Designers work on the design of recyclable plastic packaging while meeting the conditions of these functional demands. On the basis of the above cases, it can be concluded that the design for recycling plastic packaging mainly involves material design, structure design, and decoration design.

(1)Material design

It is known that the separation and recycling of commercial multilayer packaging are of great difficulty [77,78]. Currently, the substitution of multilayer packaging by mono-material packaging is possible due to the development of new materials and technologies. For example, in 2019, Colgate developed the world’s first recyclable toothpaste tube, which was certified by the APR [79]. The traditional combination of low-density polyethylene (LDPE) and aluminum is replaced by different grades of HDPE with different thicknesses, which can be correctly classified by materials recovery facilities (MRFs) in America. Guerritore et al. prepared a novel mono-material flexible film with high barrier properties by applying graphene oxide (GO) and graphene oxide/montmorillonite (GO/MMT) hybrid coatings on polyolefin substrates. It was confirmed that the coated films were easy to recycle, and the existence of nanofillers did not affect the mechanical properties of recycled films, which provides a sustainable option with respect to commercial multilayer packaging [80]. Besides, bio-based plastics and biodegradable materials, such as PLA, polyhydroxyalkanoates (PHAs), and thermoplastic starch plastics, are possible substitutes for conventional plastics. Meereboer et al. have summarized the advantages and disadvantages of PHAs. Although PHAs have good biodegradable behavior in both aerobic and anaerobic conditions, the addition of certain additives and high production costs limit their application in packaging. In this case, the natural fibers and fillers are incorporated to optimize the service life properties as well as to reduce the cost [81].

(2)Structural design

The innovation of structure design is very important for the recyclability of plastic packaging, including multifunction, assembly and disassembly, and so on. For instance, traditional pumps usually consist of a metal spring, a glass or stainless steel bead, as well as a variety of plastic parts. The complicated disassembly process and high cost of recycling pumps seriously affect the enthusiasm of recycling factories. In order to solve this problem, Tianzhou Packaging attempted to invent all-plastic pumps with different structures, and one of these pumps is made from PP with a certain percentage of post-consumer recycled (PCR) plastics. Moreover, the specific elastic reset component can avoid prolonged compression of the spring and extend the service life of the pump. In 2022, Coca-Cola released a new kind of plastic bottle with attached caps. Aside from being convenient to open and close, the cap can be recycled together with the bottle instead of being discarded into the environment after consumption.

(3)Decoration design

Although colorful and luxurious packaging can attract the attention of consumers and may promote sales, the excess inks and pigments, as well as oversized labels, are unfavorable for recycling. To solve the above problems, some brands attempt to reduce or even remove the label from plastic drink bottles [82]. However, information about brand identity, product information, and anti-counterfeiting are indispensable parts of product packaging. In consequence, embossing, laser printing, and electronic tagging are adopted to improve the design of bottles and caps. Take the electronic tag as an example; consumers can obtain product information by scanning the QR code printed on the caps [83]. At present, bottled drinks without plastic labels are available in online stores. In addition, more and more companies are selecting transparent and light-colored bottles to replace opaque and dark-colored bottles.

## 3. Recycling Technology of Plastic Packaging Wastes

In the past decades, recycling of plastic wastes has always been the research emphasis and promotional focus of academia and industry. According to the Global Recycled Standard, recycling of plastic wastes is graded into four categories in priority order [84,85], as shown in Figure 2. Primary and secondary recycling is material regeneration, which is mechanical recycling [86,87]. Tertiary recycling is chemical recycling, including the production of chemicals and oils [88,89]. Quaternary recycling is about plastic waste incineration and then recovering the energy [90,91]. In view of the fact that the process of energy recovery generates poisonous gases such as hydrogen chloride, dioxin, and polycyclic aromatic hydrocarbons, the application of energy recovery heavily depends on the development of an environmentally friendly incinerator. In view of this, this section mainly summarizes the mechanical and chemical recycling of plastic packaging waste.

### 3.1. Mechanical Recycling

As the most common technology for plastic recycling, mechanical recycling includes collection and sorting, washing and drying, grinding, melting, and extrusion [92,93,94,95], which is shown in Figure 3. Note that there is no change to the chemical composition of plastic waste in the above process. Normally, the components of plastic waste are complex and may contain different varieties of plastic, rubber, metal, and organic contaminants [96]. Due to the different processing properties, each type of plastic has to be sorted first before recycling, which is based on shape, density, size, color, or chemical composition.

#### 3.1.1. Automatic Sorting Methods

The frequently used automatic sorting methods are as follows [97,98]: air sorting, flotation and froth flotation, melt filtration, NIR, and X-ray sorting.

(1)Air sorting

The flakes of mixed plastics are fed vertically into the air first, and then the light pieces and heavy fragments are separated from each other because of the difference in specific gravity. For example, air sorting has been widely applied to the separation of PET flakes and label films. The separation rate of air sorting largely depends on the density difference, wind speed, airflow inclination angle, and height of the separating zone.

(2)Flotation and froth flotation

Flotation is also known as the sink–float method, which is related to the density and surface energy of plastic flakes [99]. Water, saturated salt solutions, alcohol solutions, and other kinds of liquids with different densities can be used as floating agents. It is known that the densities of PP and PE are lower than that of water (1 g/cm^3^), while the density of PET is higher than 1 g/cm^3^. Therefore, PET flakes can be selectively separated from PET/PE or PET/PP mixtures using water as a floating agent. But flotation sorting technique is inapplicable to separate plastic mixtures having density overlaps.

Froth flotation is a useful technique to separate plastic mixtures with similar densities, which can be understood as a combination of flotation and surface treatment. The hydrophilic or hydrophobic properties of the specific plastic particles in mixtures are changed by adding a selected wetting agent. These modified plastic particles may sink to the bottom or float on the surface before separating from the unmodified plastic particles. For instance, Saisinchai [100] found that it is possible to separate PVC from PET using Calsium Lignosulfonate as a surfactant for PET and pine oil as a frothing agent, and the recovery rate of PVC is possible to achieve 100% through regulating the component ratio of the PET/PVC mixture, concentration of wetting reagent, and the number of cleaning flotation. Nowadays, froth flotation is still in the development stage and has not been widely applied to plastic sorting.

(3)Melt filtration

Contaminations such as paper, wood, sand, and rubber particles have a negative impact on the quality and properties of recycled plastics. Melt filtration is an effective approach to separate non-melting contaminations and polymer particles with high melting points from the melt [101,102]. Melt filters have different mesh sizes, and a smaller mesh size can remove more contamination, leading to improved process stability and the mechanical properties of recycled plastics. It is noted that the presence of contaminants may cause filter blockage, which then results in pressure fluctuation.

(4)NIR and X-rays

The NIR sorting technique is widely used to identify the type of plastic; the reasons are that different plastics reflect unique spectra under near-infrared light and NIR has a high speed of identification [103,104]. But the accuracy of NIR can be affected by sample thickness, surface contaminations, and the existence of labels. For example, if the sensor detects a label rather than a plastic bottle, NIR will provide a false result of identification. In addition, as an optical surface technique, NIR is not suitable for multilayer plastic films or dark-colored plastics.

Although the X-ray sorting technique is not able to identify polymer types, it is very suitable for the identification of PVC [97]. When exposed to X-rays, the chlorine atoms in PVC present a unique peak that is easily detectable. Moreover, X-rays can be used for the identification of dark plastics and contaminants on the plastic surface.

#### 3.1.2. Individual Plastics

Another challenge for mechanical recycling of plastic waste is the degradation caused by certain conditions [105,106]. For one thing, plastic products are usually exposed to heat, oxygen, light, and mechanical stress during their lifetime, which results in photo-oxidation. For another, the reprocessing of recycled plastics induces thermal-mechanical degradation. The low-molecular volatile compounds produced by degradations probably corrode the processing equipment and reduce the performance of recycled plastics. In this case, adding additives like heat stabilizers, compatibilizers, and fillers are good options for improving the recyclability and properties of recycled plastics [107,108,109].

It is known that the recycling of PVC is more difficult than that of other general plastics, due to its poor thermal stability and complex composition. PVC may undergo severe degradation under high temperature conditions, and the produced hydrochloric acid gas (HCl) can accelerate its thermal decomposition process, while its dangerous degradation products cause serious corrosion to processing equipment [110]. Thus, thermal stabilizer is one of the most important additives for PVC, such as Ca/Zn and sulfur organotins. Asawakosinchai et al. [111] investigated the effect of organic heat stabilizers on the recycling ability, mechanical thermal stability, and mechanical properties of PVC. It was found that PVC stabilized with uracil (DAU) and eugenol exhibited excellent short-term thermal stability, and its color did not change within 3 processing cycles. However, the recycled PVC is not able to produce recycled products directly because of its poor performance and low applicability. As a result, the recycled PVC is required to blend with virgin PVC and/or with other proper thermoplastics to produce qualified products [112].

Although the thermal degradation of PE, PP, and PET is not so serious, the formed molecular defects still have a bad effect on their mechanical properties [113,114,115]. Colucci et al. [116] found that the elastic modulus, tensile, and flexural strengths of recycled PP were obviously lower than those of virgin PP, and glass fibers were adopted to reinforce the recycled PP. The mechanical properties of recycled PP composites largely depend on the changing length of glass fibers during the injection molding process, and the prepared recycled PP composites were successfully used for producing new automotive components. Thumsorn et al. [117] reported that the addition of ammonium polyphosphate (APP) and inorganic fillers significantly improved the flame retardancy and mechanical performance of recycled PET due to the synergistic effect of talc, glass beads, and APP. Besides, carbon nanotubes (CNTs) provide good reinforcement for recycled PET [118]. It is found that CNTs have heterogenous nucleation on the crystallization of recycled PET and increase the crystallization temperature and degree of crystallinity of recycled PET. The combination of an increased degree of crystallinity in recycled PET and interactions between CNTs results in increased viscosity, thermal stability, and mechanical properties of recycled PET.

#### 3.1.3. Mixed Plastics Waste

In the recycling process of mixed plastics, not every kind of plastic can be completely separated, such as composites, multilayers, or PP/PE blends. Therefore, the treatment of mixed plastics is an important developing trend in mechanical recycling [119,120]. A plastic blend system generally consists of two or more different plastics, and the miscibility between different polymer components determines the resulting performance of recycled plastic blends [121,122]. As reported, most plastic blends of different chemical structures are proven to be immiscible, leading to the formation of multiphase [123,124,125]. Techawinyutham et al. [126] investigated the mechanical, thermal, and rheological properties of recycled LDPE/PETG blends and HDPE/PETG blends without compatibilizers and found that the comprehensive properties of recycled polymer blends were lower than those of the neat recycled polymers.

The immiscibility of polymer blends can be improved by proper modification techniques, and the most common, effective, and convenient of these is the addition of compatibilizers, which promote the formation of physical or chemical bonds between immiscible polymeric components [108,127]. The specific chemical reaction or thermodynamic interaction between compatibilizers and polymers can reduce the interfacial tension, increase the thickness of the interface layer, and decrease the size of dispersed phase particles, resulting in a reduction of the interfacial tension coefficient and the formation and stabilization of the desired morphology [124,128].

In general, compatibilization is divided into physical and reactive compatibilization [129]. Physical compatibilization is the insertion of a block or graft copolymer with certain segments that are miscible with polymeric components and mainly concentrate in the interphase. For example, in order to recycle dual plastic wastes (PET bottles and PE bags), poly (ethylene-co-methacrylic acid) copolymer (EMAA) was adopted as a compatibilizer to process PET/PE blends [130]. Although recycled PE and recycled PET are not compatible, the non-polar segments (ethylene) and polar segments (methacrylic acid) of EMAA react with recycled PE and recycled PET, respectively; therefore, these two materials can form a compatible system (as shown in Figure 4). It is confirmed that the addition of EMAA improves the mechanical and thermal properties of recycled PET/PE blends (mass ratio: 3:1), and the optimum content of EMMA is 3 wt%. However, the addition of block or graft copolymers may form micelles, which lower the efficiency of the compatibilizer and then weaken the mechanical properties of polymeric blends.

Reactive compatibilization is based on the chemical reaction between two polymeric components, and the graft or block copolymers form in situ during mechanical blending [131,132]. The generated covalent or ionic bonds link the immiscible polymeric components, which decrease the size of the dispersed phase and reduce the interfacial tension. Ahmadlouydarab et al. [133] investigated the effect of PP-g-MA on the morphological and mechanical properties of r-PET/PP blends. Owing to the addition of PP-g-MA, the size of r-PET particles becomes smaller and distributes in the PP matrix uniformly. Finally, the elastic modulus, yield stress, and impact energy of r-PET/PP (10/90) blends were markedly improved, and the optimal content of PP-g-MA was 2%. Touati et al. [134] used a twin-screw co-rotating extruder to prepare PP/r-LDPE blends and compatibilized PP/r-LDPE blends of different compositions, and the influence of maleic anhydride functionalized ethylene copolymer rubber (MAC), maleic anhydride functionalized ethylene copolymer rubber/SiO_2_ (MAC/SiO_2_), and maleic anhydride functionalized ethylene copolymer rubber/SiO_2_/ionic liquid (MAC/SiO_2_/IL) on the performance of PP/r-LDPE blends was studied systematically, as shown in Figure 5. In their work, it is found that the Young’s modulus, stress at break, and elongation at break of PP/r-LDPE blends were significantly improved by the three kinds of compatibilizers, and MAC/SiO_2_ (5/3) has a better compatibilization effect than that of other compatibilizers due to the synergy effect between MAC and nano-SiO_2_. Martikka et al. [135] studied the effects of different kinds of compatibilizers with different levels of addition on the properties of wood-polymer composites. It was found that the mechanical properties and moisture resistance of wood-polymer composites were increased by 50% or more through the addition of selected compatibilizers, indicating that the addition of proper compatibilizers provides a feasible way to tailor the properties of wood-polymer composites.

With the increased application of biopolymers, the recycling of biopolymer-based blends has attracted much attention [136,137,138]. However, the separation of biopolymers from petroleum-based polymers is complicated and not yet present in the waste sorting system. As a result, bioplastic waste, such as PLA, always coexists with conventional plastic waste. It is reported that PET and PLA bottles are transparent and have similar densities, and the accuracy of NIR for separating PLA from PET bottles is between 86% and 95% [139]. Moreover, even a small amount of PLA has a negative influence on the performance of PET because of the incompatibility between the two materials. As it is impossible to completely remove PLA from PET bottles, the study of improving the compatibility between PLA and PET is necessary. Gere and Czigany [140] prepared a series of r-PET/r-PLA blends using a twin-screw extruder, and the size of dispersion phase particles was dramatically decreased due to the addition of ethylene-butyl acrylate-glycidyl methacrylate (E-BA-GMA), as illustrated in Figure 6. During the processing, the epoxide group of E-BA-GMA reacts with the carboxyl and hydroxyl end groups of r-PET and r-PLA, resulting in the combination of r-PET and r-PLA chains as well as crosslinking. Although the Young’s modulus of r-PET/r-PLA blends was slightly decreased, the elongation at break, Charpy impact strength, and thermal stability of r-PET/r-PLA blends were significantly enhanced.

Because of its low cost and simple technology, mechanical recycling becomes the only large-scale application of the plastics recycling technique, but it is not suitable for plastics with poor thermal stability and thermosetting plastics. Besides, the addition of organic fillers and plastic additives is necessary to increase the performance of recycled plastics, while one or more compatibilizers are required to improve the interfacial compatibility of plastic mixtures. As a result, the complicated compositions of recycled materials make the separation and recycling of recycled products more difficult. More importantly, most of the plastic wastes are inevitably exposed to unknown contaminants after consumption, and these pollutants may still exist in the recycled plastic products. In this case, the migration of these pollutants from recycled plastic products into their contents is probably harmful to human health. Therefore, the development of novel multi-phase compatibilizers and high-efficiency recycling techniques, as well as the improvement of the standardization evaluation system of recycled plastics and their products, will be the emphasis of research in the future.

### 3.2. Chemical Recycling

Chemical recycling of plastic wastes is the use of thermochemical and catalytic conversion techniques to break down plastic wastes into smaller molecules such as monomers and oligomers [141,142,143]. The main methods of chemical recycling are pyrolysis, gasification, hydrocracking, or depolymerization. Because the chemical products of chemical recycling have similar characteristics to petrochemical products, they can be used to replace some fossil fuels. By comparison with physical recycling, chemical recycling has greater potential to recycle mixed plastic waste or contaminated plastics. However, only a small number of plastics can be converted into fuels or monomers through chemical recycling due to the limitations of current techniques and instruments [144,145]. In this part, the typical thermoplastic, thermosetting, and mixed plastics are chosen to summarize the development status of chemical recycling of plastic wastes.

#### 3.2.1. Thermoplastic Waste

As one of the most promising approaches to realizing the upcycling of plastic wastes, chemical recycling of thermoplastic waste has been developing rapidly in recent years [146,147]. Bäckström et al. [148] adapted nitric acid as an oxidizing agent to degrade PE waste into dicarboxylic acids under microwave-assisted hydrothermal conditions. It was found that the total yield of reaction products reached 71%, and the carbon efficiency of the above process was 37%. Besides, the length of dicarboxylic acids can be tuned by controlling the reaction time and temperature, as well as the content of nitric acid. In order to further realize the upcycling of PE waste, the prepared dicarboxylic acids such as succinic, glutaric, and adipic acids were utilized to prepare a value-added plasticizer with crotonate end groups, as shown in Figure 7 [149]. After the addition of plasticizer, the strain at break of grafted PLA increased from 6% to 156%, and the glass transition temperature of PLA was reduced by 10 °C, due to the plasticization and compatibilization of the grafted plasticizer. Kots et al. [150] studied the hydrogenolysis of PP waste in the presence of different transition metal/titania (M/TiO_2_) catalysts. It was confirmed that Ru/TiO_2_ was an active and selective catalyst for PP waste, and the hydrolysis of PP waste occurred at relatively low temperatures with a modest H_2_ pressure. As a result, the formation of lubricant-range hydrocarbons (66–80%) with low contents of methane was achieved, making it possible to replace the commercial base or synthetic oils.

The upcycling of aromatic plastic waste has also been reported. In the work of Sharma et al. [151], the post-consumer PET flakes were depolymerized in a microwave reactor and then converted into terephthalamides in the presence of ethylene diamine. After that, the amine-terminated terephthalamides, together with paraformaldehyde and cardanol, were used to prepare bis-benzoxazine resins containing amide linkages through Mannich-type condensation. It was found that the curing temperature of the above-mentioned synthesized resins was lower than mono-benzoxazine due to the high functionality of oxazine; moreover, the cured benzoxazine resins have high thermal stability and good lap shear strength because of the large amount of polymerizable benzoxazine. Jing et al. [152] investigated the effect of a Ru/Nb_2_O_5_ catalyst on the hydrogenolysis of various aromatic plastic wastes. The Ru/Nb_2_O_5_ catalyst is a multifunctional catalyst that can selectively break the C-O and C-C linkages of aromatic plastics and also prevent the hydrogenation of aromatic rings. As a result, the Ru/Nb_2_O_5_ catalyst not only allows the selective conversion of individual aromatic plastics but also enables the simultaneous conversion of a mixture of aromatic plastics to arenes with a high yield (75–85%), as shown in Figure 8.

#### 3.2.2. Thermosetting Plastics Waste

Thermosetting plastics exhibit better mechanical properties, thermal stability, and chemical resistance than thermoplastics because of their stable three-dimensional network structures [153,154]. However, thermosetting plastics are a kind of inmeltable and insoluble material, which increases the difficulty of recycling. At present, thermosetting plastic waste is mainly used as fillers after grinding, but the added value of recycled products is extremely low. With the increasing utilization of thermosetting plastics, there is an increasing interest in the study of chemical recycling of thermosetting plastics, especially epoxy resin and unsaturated polyester resin (UPR) [155,156].

The common methods to recycle epoxy resins contain supercritical water treatment, acid degradation, oxidative degradation, and electrochemical degradation [157,158,159,160]. For example, Kim et al. have studied the green recycling of carbon-fiber-reinforced epoxy composites (CFRP) using supercritical water without any catalyst or oxidant [157]. Through investigating the extent of degradation of epoxy resin and the surface structure and physical properties of recovered carbon fibers, they obtained the optimum degradation conditions for epoxy resin. It was found that up to 99.5% of the epoxy resin was decomposed, and the recovered carbon fibers were used to modify cyclic butylene terephthalate. As a result, a kind of conducting carbon fiber composite with high thermal and electrical conductivity was prepared. However, the degradative chemicals of epoxy resin cannot be utilized effectively due to their complex compositions. The desired recycling process for thermosetting plastic waste is to realize the high value-added application of degradation products. Varughese et al. [161] offered an effective and environment-friendly method to recycle CFRP, as shown in Figure 9. In their study, an aqueous mixture of acetic acid and hydrogen peroxide was used to treat the epoxy resin under mild conditions. After the oxidative degradation of epoxy resin, it was found that the surface structure and tensile properties of the recovered carbon fibers were close to those of the virgin fibers. Meanwhile, the recovered epoxy was confirmed to have good mechanical properties and was able to be reused through mixing with an adhesive-grade epoxy. In addition, more than 90% of the solvents could be recovered by means of a simple distillation process, demonstrating that the one-step oxidative process of the recycling of CFRP is an upgraded and sustainable route.

It is reported that the conventional chemical recycling of UPR is focused on the decomposition of the resin by solvolysis and is designed to produce monomers or oligomers. Deng et al. [162] developed a selective method to recycle oligomers and monomers from UPR, as well as glass fibers from glass fiber-reinforced UPR (GFRP). As UPR is mainly built up by C-C, C-H, and C-O bonds, the AlCl_3_/CH_3_COOH system was selected to break the C-O bond and leave the carbon skeleton intact under certain conditions. Therefore, more than 90% of monomers and oligomers were recycled from UPR; furthermore, the recovered glass fibers present similar structure and tensile strength compared with virgin glass fibers. In recent years, Wang and his group [163,164] proposed an effective and facile approach to produce high-value products through the chemical recycling of UPR under mild conditions, as illustrated in Figure 10. The binary alkalis diethylenetriamine and sodium hydroxide were adopted for the hydrolysis of UPR, and the compositions of the degraded products can be regulated by changing the content of the binary alkalis. Significantly, a type of UP-gel containing active groups such as carboxylate and amine groups was obtained and exhibited excellent adsorption capacities for cationic dyes and heavy metal ions. It is worth noting that the function of these products is largely dependent on the reaction conditions and the composition of UPR waste.

Chemical recycling is applicable for the recycling of thermosetting plastics and plastic mixtures and is recommended as a complementary solution to mechanical recycling. Chemical recycling has various methods, good designability, and controllable products; therefore, it has great potential for plastic upcycling. Nevertheless, there are still many scientific and technical problems that need to be solved before scaled applications can be realized, such as strict operation conditions, the high cost and poor reusability of catalysts, and complicated purification processes. In future research, more effort needs to be dedicated to the study of the deactivation mechanisms of catalysts, the development of high-efficiency and low-cost catalyst systems, the design and research of pyrolysis instruments, as well as the optimization of purifying technology.

## 4. Conclusions

At present, plastic waste management has become the second-biggest environmental problem after climate change and has brought a great challenge to global sustainable development. Although governments and environmental groups attach great importance to the treatment of plastic waste at the policy level, the control effect is far from satisfactory. Thus, it is essential for the plastics industry to change from a linear economy model (take-make-dispose) to a circular economy. From the perspective of the life cycle of plastic packaging, the present work emphasized the importance of combining the front-end design for recycling with the back-end recycling technologies, which not only can reduce the amount of plastic waste but also can improve the quality of recycled plastics.

The design for recycling plastic packaging can be implemented through the design and selection of materials, structure design, and decoration design. Firstly, mono-material plastic or degradable materials are preferentially used for plastic packaging. But if the properties of individual plastics cannot meet actual requirements, multi-component materials having good compatibility or easy separability are also under consideration. Secondly, assemble-and-disassemble structure designs are beneficial to simplify the sorting and recycling of plastic waste. In addition, an alternative structure design for multi-layer packaging is a positive attempt to solve the problems of separation and recycling. Thirdly, it is necessary to simplify the decoration design of plastic packaging under the premise of satisfying the requirement of information integrity. That is, excessive inks or coatings, oversized labels, and unnecessary attachments need to be removed or optimized.

As the current recycling techniques have their advantages and disadvantages, it is hard to achieve the purposes of high performance, easy processing, and low cost at the same time. Thus, there is an urgent need to develop new and efficient recycling techniques, such as multi-phase compatibilizers, green and highly effective catalysts, and pyrolysis recycling methods with low energy consumption. Multifunctional sorting and recycling machines also have important functions in increasing the recycling efficiency and quality of recycled products. Meanwhile, the variety identification of plastic wastes and the quality evaluation of recycled plastics need to be gradually improved, which is beneficial to standardizing and expanding the recycled plastics market. Besides, it is suggested that the recycling enterprises should keep in touch with manufacturing companies. In this way, to solve the recycling problems caused by improper design, the latest developments in recycling equipment and techniques are able to provide feedback to the design process in real time, which can encourage designers to improve the current design.

## Figures and Tables

**Figure 1 polymers-15-01485-f001:**
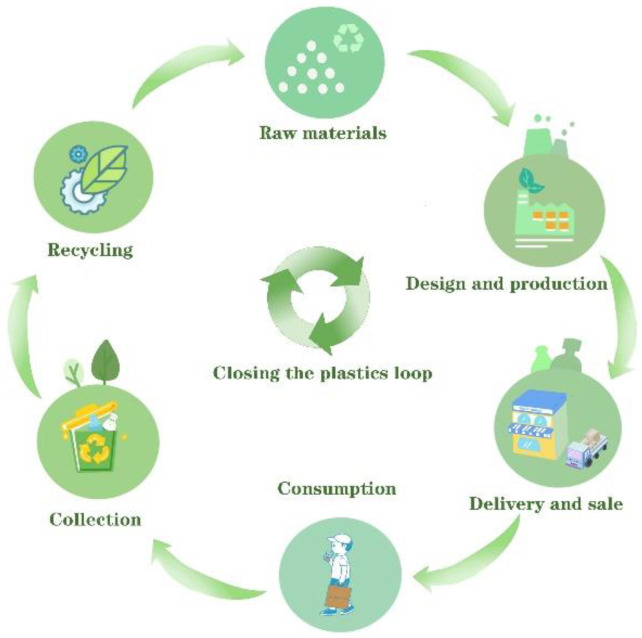
Schematic diagram of the closed-loop circulation of plastics.

**Figure 2 polymers-15-01485-f002:**
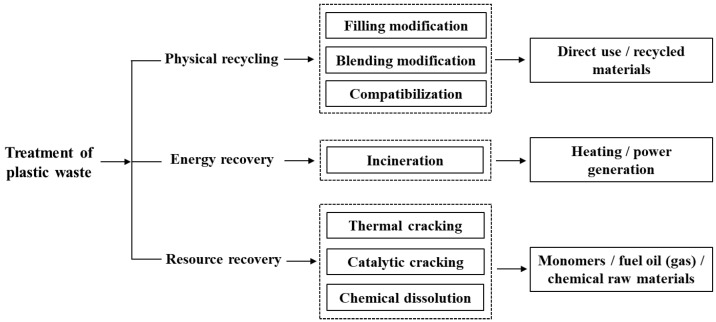
Current recycling technologies for plastic waste.

**Figure 3 polymers-15-01485-f003:**
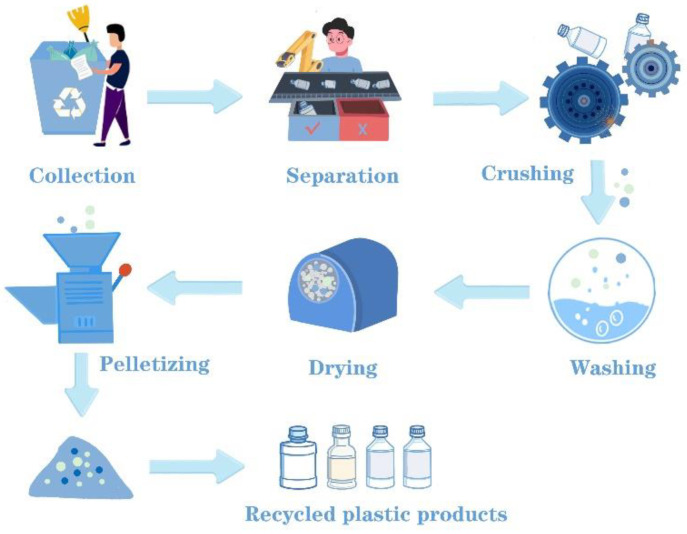
Process of mechanical recycling of plastic wastes.

**Figure 4 polymers-15-01485-f004:**
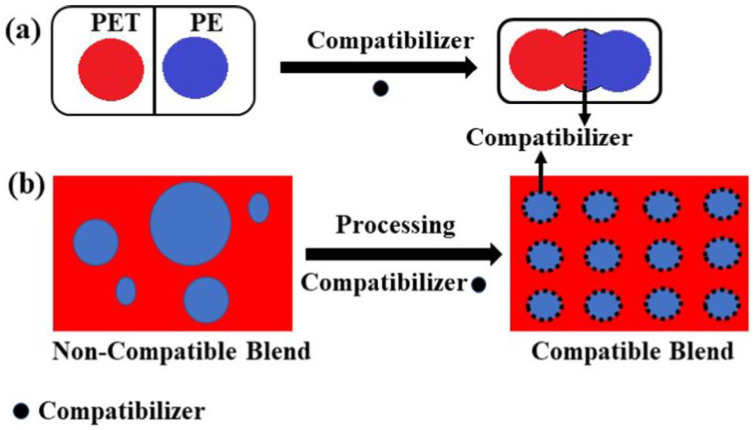
A schematic representation of the role of the compatibilizer. (**a**) Function of compatibilizer unit. (**b**) Effect of compatibilizer in processing of blend. Reprinted with permission from Ref. [130].

**Figure 5 polymers-15-01485-f005:**
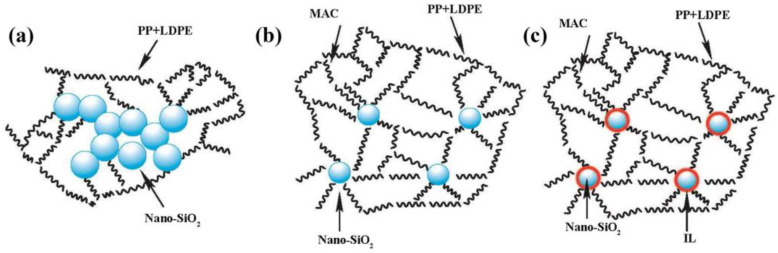
A schematic representation of the effects of (**a**) MAC, (**b**) MAC/SiO_2_, and (**c**) MAC/SiO_2_/ionic liquid on PP/r-LDPE blends. Reprinted with permission from Ref. [134].

**Figure 6 polymers-15-01485-f006:**
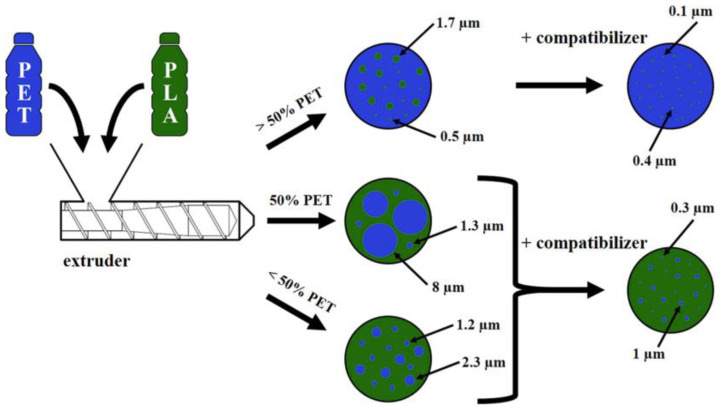
Dispersed phase structures of different uncompatibilized and compatibilized PET/PLA blends. Reprinted with permission from Ref. [140].

**Figure 7 polymers-15-01485-f007:**
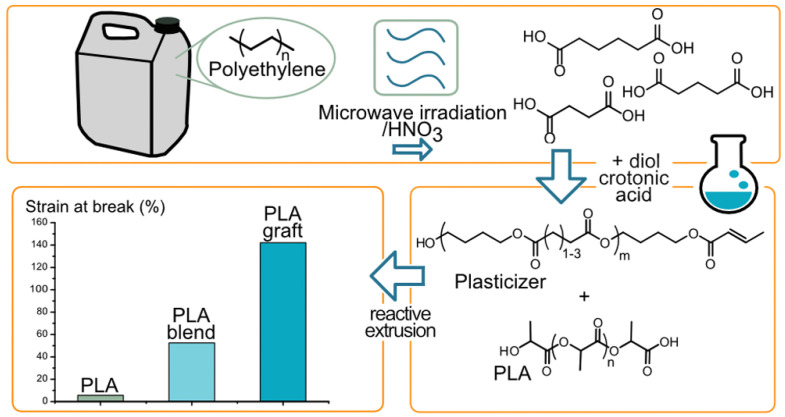
HDPE upcycling to telechelic carboxylic acid via HNO_3_ oxidation and its further application to synthesize plasticizer within PLA. Reprinted with permission from Ref. [149].

**Figure 8 polymers-15-01485-f008:**
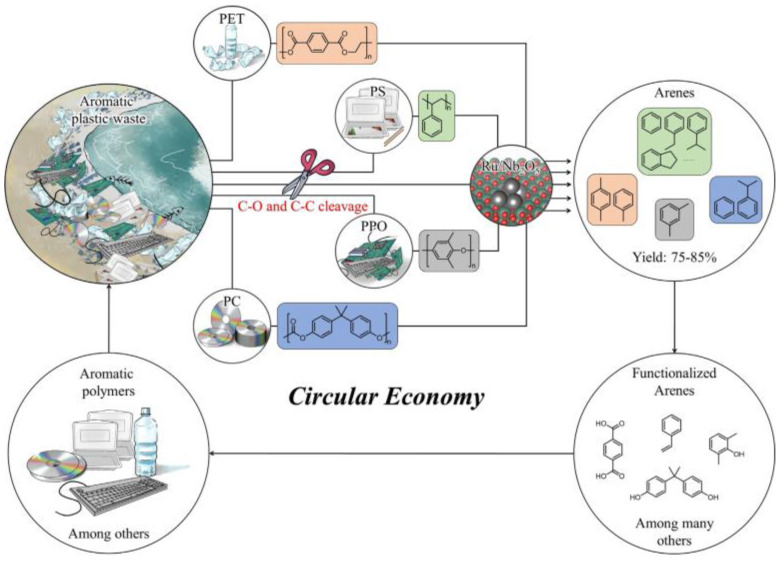
The integration of a C-O and C-C bond cleavage catalyst into the circular plastic economy. Ru/Nb_2_O_5_ is proposed as a possible catalyst for the conversion of various aromatic plastic wastes. Reprinted with permission from Ref. [152].

**Figure 9 polymers-15-01485-f009:**
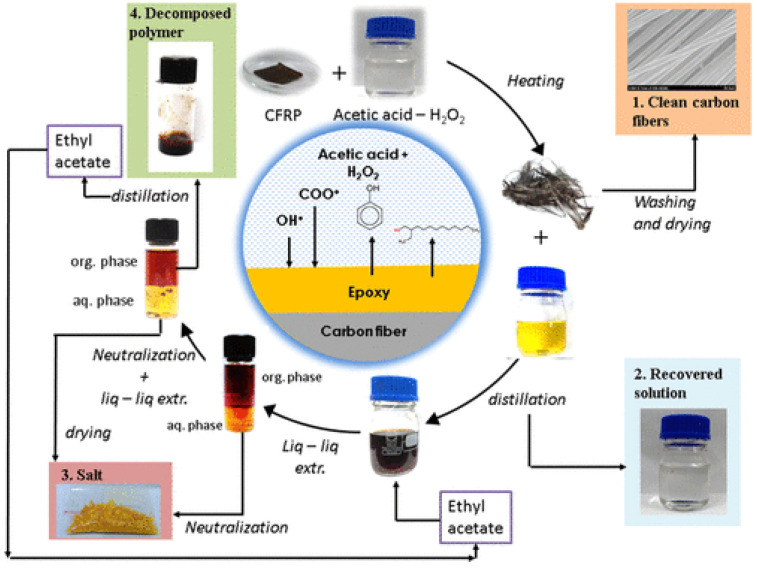
Schematic diagram of the green recycling process of CFRP. Reprinted with permission from Reference [161].

**Figure 10 polymers-15-01485-f010:**
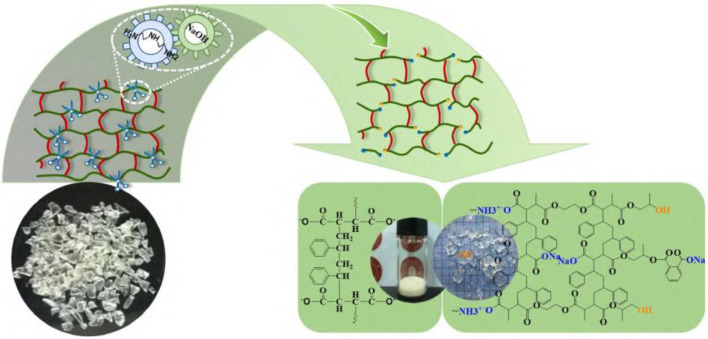
Selective hydrolysis of ester bonds and controlled conversion of UPR into gel or poly(styrene-maleic acid). Reprinted with permission from Reference [163].

**Table 1 polymers-15-01485-t001:** Application cases of design for recycling of plastic packaging.

Component	Original Design	Improved Design	Packaging Form	Applicable Products	Brand Name
Barrier layer	aluminized PET multilayer composites	Mono-polyolefin HDPE * PET PP	soft tube blister pack skin packaging packaging film	tooth paste cosmetics cheese fruit paste vegetable paste milk powder	Nestle Colgate Essel Gerber
Adhesive	solvent based/ pressure-sensitive/melt adhesive	washable adhesive self-adhesive non-glue structure	bottle bucket can flexible package express package	daily chemical/beauty products drinks/milk machine oil commodity	UPM Raflatac Avery Dennison China Post Amazon
Label	PVC shrink label paper label multi-layer label	PET shrink label PETG * shrink label wood-based PE Peelable label No label label reduction Laser printing and embossing Electronic tag	bottle container flexible pack	Mouth wash coffee/tea/juice carbonated/ isotonic drink gum	ORION Darlie Master Kong Eastroc Beverage Coca Cola Pepsi Pulpy Orange Evian
Pump	metal spring metal/glass bead PP/PE	All-plastic pump 100% PP 100% PE PP/r-PP	bottle container	daily chemical/ cleaning products medicine skin care	Tianzhou Aptar Group Berry Global Rieke Packaging
Color	green/blue white/black opaque	transparent unpigmented	bottle	coffee/juice carbonated drink milk/tea daily chemical skin care	Spite Fido Amcor Unilever Cnnice
Attachment	separated cap aluminum foil	attached cap EVA * TPE *	bottle soft bag	drinks sauce	Alpla Sidel Coca Cola

* HDPE, PETG, EVA, and TPE are the abbreviations for high-density polyethylene, polyethylene terephthalate G copolyester, ethylene vinyl acetate copolymers, and thermoplastic elastomers, respectively.

## Data Availability

Not applicable.

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
