# Peer review of "The Key to Solving Plastic Packaging Wastes: Design for Recycling and Recycling Technology"

_polymers, 2023, doi:10.3390/polym15061485_

Round 1

Reviewer 1 Report

This paper reviews the different designs for recycling and recycling technologies. The review may be of interest to the scientific community, but should be completed.

The work would be more complete if a section were included with the researches that are being carried out in this field. The innovative strategies that some brands are following are mentioned, but the advances that are being made today in science are not presented in detail. In this sense, there is no talk of plastic mono-materials, bio-materials or PHA, for example.

On the other hand, it could be completed with the regulation and the measures taken. These are mentioned but not delved into.

It would also be interesting to have comparative values of recycling and the state of the art in terms of technology in different countries or continents.

These points could make the work complete and suitable for publication.

Specific Comments:

- Table 1. The table would provide more information if the corresponding information were specified for each component for each brand name or application of the product.

- Acronyms must be defined the first time they are used: As has been done for PETG, include in the text or where it is used, the definition of each abbreviation (EVA, TPE, etc.).

Reviewer 2 Report

The key to solving plastic packaging pollution: Design for recycling and recycling technology

The article covers the design and methodologies used to improve plastic recycling systems. It covers the history and future ideas as well as policies around plastic recycling.

It is well-written article with relevant information from recently published articles. Here are my comments:

Line 29-30: Please rephrase the sentence to deliver the intended message

Line 86 please remove ‘%’ from the value 80

Line 107-109: Please rephrase the sentence to deliver the intended message

Line 118-119: insert ‘be’ between ‘to’ and ‘reused directly’

Line 290-219: Please rephrase the sentence to deliver the intended message

Line 489-492: Please shorten the sentence so that it can deliver the intended message.

Line 501-504: Please shorten the sentence so that it can deliver the intended message.

Line 629: what kind of packaging? Since the recycled plastics are often contaminated hence cannot be used for other forms of packaging, e.g. food packaging or biomedical packaging.

Reviewer 3 Report

Authors are kindly invited to remove the systematic use of the tendentious "plastic pollution" all along the manuscript, changing it by "bad management of plastic wastes", or just "plastic wastes" at the best convenience.

The 2.2. Sub-Section entitled "Guidence and Inlfluencing factors" must be removed at all in order to avoid tendentiousness and indoctrination derived from the absence of scientific content, but lack of cost effectiveness, in a problem with satisfactory technical-scientific solutions from at least fifteen years ago.

In such sense and under the properly named review Sections, the PET recycling processes as described ought to be improved. Authors seem unknown, for example, that the so claimed as "Non-recyclable at all", PET, at early eighties on last Century, became recyclable and it was so claimed by Dupont, just once the US, FDA forbid his application as container into the carbonic beverage markets. In the same sense, and when mentioning the hazards derived from the Cl presence in plastic waste fractions containing PVC and its recycling processes, any mention to the Vynil-Loop processes is missing. Authors do not make any mention regarding risks derived from true pollution from external sources able to survive into the bulk polymers along the so-called mechanical recycling processes, in those applications in food, or skin contact, for example.

Authors are kindly invited to clarify the role of Standardization when the recycled polymers re-enter in the industrial application markets by just matching the requirements of the performance that those go destined.

Under the properly entitled Sections and sub-sections devoted to plastic recycling, more that truism comments about one or several references, a sharp description of the different works compiled and reviewed by authors would be that one should expect in a claimed as review manuscript. In such sense, comments by authors under the chemical recycling processes of thermoset reinforced composites contain main reasons illustrating the hard cost effectiveness, and not technical problems which command the recycling.

How the balance between the raw materials producers markets and the molded parts and compounding maker ones,  affect the recyclability in terms of circular economy in a context of free market, invites to a serious reflection by authors in order to re-write a fully new "4. Conclusion" Section. Any comment of politic nature and, or, economic guidelines are outwards the aim and scope of a Journal devoted to the Polymer Field and must be removed.

Reviewer 4 Report

Reviewer Report

Comments for Authors

Dear authors, I have read your review paper The key to solving plastic packaging pollution: Design for recycling and recycling technology my comments and suggestions are mentioned below:

1.      The present review paper is well written and organized.

2.      The general structure and flow is clear for the readers and relevant for solving plastic packaging pollution.

3.      The present review paper presents solutions for: (i) Design for recycling of plastic packaging; and (ii) Recycling technology of plastic packaging waste.

4.      Figures of the review paper are adequate for presented subject.

5.      The conclusions are relevant for presented subject in this review paper.

In my opinion, the review paper is suitable for publication.

Author Response

Thank you very much for your comments!

Round 2

Reviewer 1 Report

Practically all the comments have been well justified and introduced into the paper. The paper is suitable for publication.

Author Response

Thanks very much for your comments!

Reviewer 3 Report

Congratulations to authors by the excellent revision work made in this manuscript!

Only two minimal corrections:

1.- page nr.2/at the beginning of the 2nd paragraph, please, change "Plastic wastes…"  by "The best management of Plastic Wastes…"

2.- page nr. 19/Under Section 4. Conclusions, 1st sentence, please, change "plastic pollution" by "plastic wastes management"
